# Effects of Hiking-Dependent Walking Speeds and Slopes on Spatiotemporal Gait Parameters and Ground Reaction Forces: A Treadmill-Based Analysis in Healthy Young Adults

**Ioannis Kafetzakis** [1], **Ilias Konstantinou** [2] **and Dimitris Mandalidis** [1,*]

1   Sports Physical Therapy Laboratory, Department of Physical Education and Sports Science, School of Physical Education and Sports Science, National and Kapodistrian University of Athens, Ethnikis Antistasis 41, 17237 Athens, Greece; kafetzos@phed.uoa.gr
2   Research and Development Department, Serinth Ltd., Smyrnis 52, 14341 Athens, Greece; think.konstant@gmail.com
*   Correspondence: dmndldis@phed.uoa.gr; Tel.: +30-69-7499-1457

**Abstract:** Hiking offers both recreational enjoyment and physical challenges, requiring speed adjustments when traversing uphill and downhill slopes. These adjustments prompt compensatory responses in kinematics and kinetics to mitigate fatigue and musculoskeletal strains. The study aimed to explore the impact of slope-specific walking speeds on spatiotemporal gait parameters, vertical ground reaction forces (vGRFs), and position of the center of pressure (COP) during uphill and downhill walking. Thirty-two healthy individuals completed five 4-min walks on an instrumented treadmill set to 0% (level), +10%, and +20% (uphill), and −10% and −20% (downhill), slopes, at 5.0, 3.5, 2.5, 5.0 and 3.5 km h$^{-1}$, respectively. Uphill walking led to reduced stride length and cadence, increased foot rotation, step time, and durations of stance, swing, and double-stance phases. Conversely, downhill walking exhibited decreased step length, step time, and durations of stance, swing, and double-stance phases but increased step width and cadence compared to level walking. Speed adjustments to accommodate slope led to reduced vGRFs for uphill and downhill walking. Additionally, the COP shifted forward during uphill and backward during downhill walking and displaced laterally as walking became more demanding. The observed responses indicate adaptations aimed at maintaining postural control, reducing excessive load application, and optimizing energy expenditure on sloping terrain.

**Keywords:** inclined walking; Tobler's hiking function; path selection; gait analysis; postural responses

## 1. Introduction

Hiking, as well as other long-distance walking activities (e.g., trekking, pilgrimage), are physically challenging recreational activities conducted in natural environments, often on trails or paths in the countryside, mountains, or forests. They can range from short, leisurely walks on well-maintained trails to more challenging and strenuous treks in remote or mountainous areas [1]. In recent years, the interest in hiking has seen a dramatic increase worldwide. In 2022, countries such as the United States witnessed the participation of nearly 60 million people in hiking activities. This marked the highest recorded participation rate in the country since 2010, reflecting a remarkable increase of approximately 83.2% (www. statista.com; accessed on 17 January 2024). The physical challenges encountered during these activities stem from the necessity to navigate diverse landscapes in outdoor settings, manifested in overcoming gravitational resistance during uphill walking and controlling excessive impact forces during downhill walking. The motivations driving many healthy individuals to face the physical challenges associated with hiking, such as the heightened cardiorespiratory effort required to overcome the stress of uphill walking [2], primarily arise from the benefits that enable them to enhance cardiopulmonary function [3,4], decrease

lean mass, mainly in the arms and legs [5], reduce the risk of type 2 diabetes [6], and improve the endurance of muscles involved in uphill walking [7,8]. Additionally, these activities contribute to improving their mental well-being [9,10]. On the other hand, the reduced intensity of movement during downhill walking [11] can help patients with COPD, diabetes, obesity, heart problems, and osteoarthritis [6,12,13] to exercise under conditions that allow them to manage health problems associated with their diseases.

Given the impact of heightened propulsive forces required to overcome during uphill walking and the effective deceleration strategies that must be implemented for downhill walking, successful and effective participation in hiking depends on adjusting one's speed to the slope of the terrain. This adaptation is expected to directly impact the spatiotemporal parameters of gait, including step length and width, step time, and cadence, and induce significant changes in vertical ground reaction forces (vGRFs) and joint moments. In this context, several researchers have explored the impact of uphill and/or downhill walking on various aspects of gait kinematics and kinetics. The kinematics of gait, in terms of spatiotemporal parameters, has been investigated by several studies, either during overground walking on ramps [14] or treadmill-based walking [15–17]. In these studies, participants walked at self-selected speeds on surfaces with slopes varying from 0% to $\pm$39% [14–19]. On similar terrain gradients, other studies employing force platforms embedded in the surface of a ramp [14,19,20] or adapted on treadmills [15] have concentrated on investigating the impacts of uphill and downhill walking at preferred speeds on vGRFs [14,15,19,20]. Additionally, some researchers, utilizing insoles [21] and sensor matrices [22], have investigated the distribution of plantar pressures under similar experimental conditions. However, despite the valuable information offered by the studies implementing self-selected speeds that replicate everyday walking conditions, promoting more natural movement patterns and reducing stress, their applicability to the general population may be limited due to potential individual variations and responses to walking conditions. On the other hand, when utilizing predetermined speeds, controlled conditions are imposed, enabling researchers to standardize and manipulate variables for a specific population and experimental design. In this regard, only a few studies have explored kinematic variables of sloping gait, either individually or in combination with kinetic variables. However, none of these studies have incorporated speeds specific to hiking, providing a more comprehensive approach to understanding the spatiotemporal and dynamic correlates of sloped walking.

Waldo R. Tobler, a renowned American–Swiss geographer and cartographer, formulated an exponential function, commonly referred to as Tobler's hiking function, based on which the speed of travel increases non-linearly with steeper slopes [23]. This function was designed to estimate the time it takes for an individual to traverse a given terrain based on factors such as slope, considering the principle that walkers tend to choose paths that minimize both uphill gradients and overall travel time [23].

$$W = 6e^{-3.5|S+0.5|}$$

where *W* = walking speed and *S* = slope of the terrain

Knowing the effects of hiking speeds and terrain's slopes interplay on gait kinematics, and kinetics is essential for promoting safety [24,25], optimizing performance, designing appropriate trails [26], and equipment (e.g., footwear [27]), enhancing recreation and fitness experiences, supporting rehabilitation [28,29], and minimizing environmental impact [30]. This information may also be valuable to individuals who, either because they face limited access to physical environments or want to diversify their exercise routines, prefer to exercise in controlled indoor conditions using a treadmill. Although they cannot fully replicate the complex challenges posed by natural terrains, treadmills offer a unique platform for individuals to reap the benefits of outdoor walking by adapting and optimizing specific aspects of it indoors [31–33]. This study, therefore, aimed to investigate how spatiotemporal parameters and vGRFs are affected by hiking-dependent walking speeds and slopes as determined by Tobler's hiking function using an instrumented treadmill.

## 2. Materials and Methods

### 2.1. Participants

Thirty-two healthy, physically active collegiate students (9 males and 23 females, age of 23.4 ± 4.1 y, height: 1.7 ± 0.1 m, body weight: 64.9 ± 11.4 kg and BMI: 22.6 ± 2.4 kg m$^{-1}$). All participants were capable of naturally walking barefoot on the treadmill's belt. They were instructed to abstain from engaging in strenuous activities before reporting to the laboratory for testing. Additionally, participants were advised to wear lightweight and comfortable clothing and to maintain their gaze in a forward direction when walking. Participants were excluded from the study if they presented excessive musculoskeletal deviations such as leg length discrepancy (>0.5 cm measured with a standard measure tape) [34], scoliosis (>5° trunk rotation in Adam's test measured with a standard scoliometer) [35], and/or foot overpronation or supination (>10 score determined with the Foot Posture Index-6 [36]). They were also excluded when they demonstrated an inability to fully bear their body weight or limping while walking, reported feeling pain, or had a medical history of neurological, visual, vestibular, or balance disorders affecting gait. Participants who expressed fatigue or discomfort while performing the study protocol were also excused from the study. Moreover, the test was interrupted, and the participant was dismissed when his/her heart rate (HR) exceeded 60% of the maximum HR [HRmax = (220 − participant's age) × 0.6] and reached 17 points on Borg's 15-point (6 to 20) rating scale for perceived exertion (PE) [37]. These thresholds have been linked to the loss of postural control [38] and significant exertion [39], which could potentially impact the participants' ability to walk normally, especially in challenging uphill and downhill conditions. Heart rate was recorded using a heart rate sensor (Polar Electro, H10, Kempele, Finland) attached to a Polar heart rate chest strap positioned at the level of each participant's xiphoid process. The heart rates detected by the sensor were wirelessly displayed on the treadmill's monitor, as the heart rate sensor was compatible with the instrumented treadmill. Both HR and PE were recorded before the commencement and at the conclusion of each gait condition. Ultimately, the average HR of the participants surpassed 60% of their HRmax solely upon completing the uphill walking condition at a slope of 20%. However, perceived exertion did not exceed 17 points at the conclusion of any of the walking conditions. Each of the selected volunteers was briefed on the study's objectives and provided written consent before participating.

### 2.2. Instrumentation

An instrumented treadmill (Pluto® Med, h/p/cosmos® Sports & Medical GmbH, Nussdorf–Traunstein, Germany) with a running/walking surface with dimensions of 150 cm (L) × 50 cm (W) and an embedded force platform of capacitive-pressure sensors (FDM-THPL-M-3i, Zebris Medical GmbH, Isny, Germany) beneath treadmill's belt was used for the implementation of gait protocol. The pressure platform's sensor area measured L: 108.4 × W: 47.4 cm and comprised 7168 sensors, collecting data at a sampling rate of 240 Hz. The sensors' threshold was preset by the manufacturer at 1 N/cm$^2$. The instrumented treadmill was equipped with features enabling a speed range of up to 18.0 km/h for uphill walking and up to 5.0 km/h for downhill walking. Its surface allowed for setting slopes ranging from 0.1% to 20.0%. The treadmill belt's forward motion facilitated walking on both level and uphill slopes (yellow arrow), while its reverse direction (blue arrow) enabled walking downhill (Figure 1). The treadmill was connected to a desktop computer, enabling real-time data transfer as well as storage for subsequent processing and analysis. Additionally, the manufacturer-provided software allowed for remote adjustment and control of the parameters utilized in the research protocol.

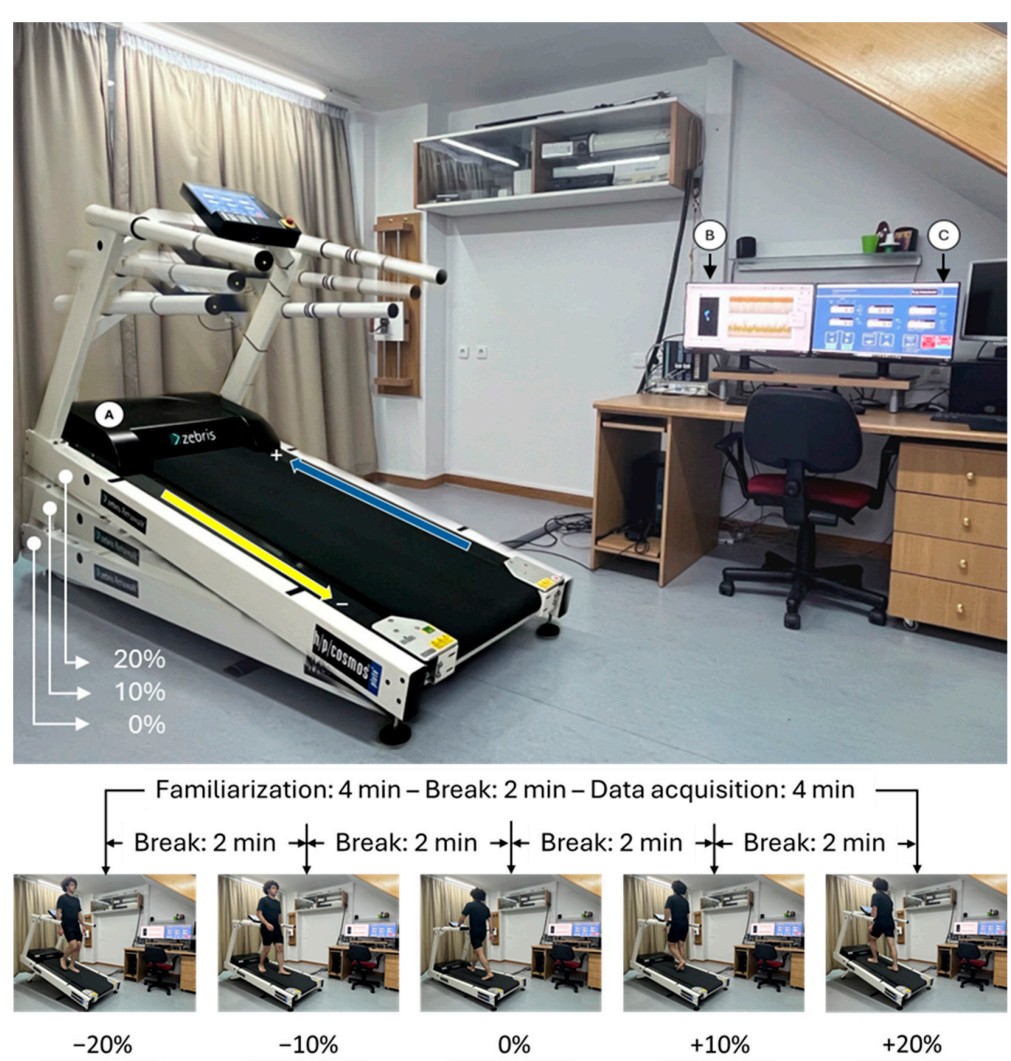

**Figure 1.** Experimental setup utilized in the research protocol: The treadmill's surface (A) configured for level walking (0%) with 5.0 km h$^{-1}$, uphill walking on +10% and +20% slopes with 3.5 km h$^{-1}$, and 2.5 km h$^{-1}$, respectively, and downhill walking on −10% and −20% slopes with 5.0 km h$^{-1}$ and 3.5 km h$^{-1}$, respectively. Display on the screen of the data transferred to the desktop (B) and the software provided by the manufacturer (C) for remote control of the treadmill.

### 2.3. Testing Protocol

Each participant was instructed to walk barefoot on the instrumented treadmill at 0% slope (level) with a speed of 5.0 km h$^{-1}$ (1.39 m s$^{-1}$), uphill slopes at +10% and +20% with speeds of 3.5 km h$^{-1}$ (0.97 m s$^{-1}$) and 2.5 km h$^{-1}$ (0.69 m s$^{-1}$), respectively, and downhill slopes at −10% and −20% with speeds of 5.0 km h$^{-1}$ (1.39 m s$^{-1}$) and 3.5 km h$^{-1}$ (0.97 m s$^{-1}$), respectively. Each walking condition required participants to undergo a 4-min walking session at the predetermined speed/slope to familiarize themselves with each walking condition, followed by an additional 4-min walking session dedicated to data acquisition [40]. To prevent fatigue, a 2-min break was provided between the familiarization and data acquisition sessions. Additionally, an extra 4-min interval was incorporated between testing different walking conditions. Slopes were selected based on terrain inclinations that are usually encountered in most urban areas [41,42] or hiking-related trails [43]. The walking speed for each slope was calculated using Tobler's exponential hiking function [23,44], with 5.0 km h$^{-1}$ considered to be the average walking speed typically used by individuals aged 20–39 years during comfortable level walking [45]. To avoid the potential fatigue effect, walking conditions were performed in a random order. This was

accomplished by instructing each participant to choose a number between 1 and 120, with each of the numbers representing a sequence of walking conditions arranged in a different and random order. The 120 possible sequences of the five walking conditions were created using a web application (https://www.random.org, accessed on 16 January 2022).

### 2.4. Data Analysis

Gait analysis was conducted, encompassing spatiotemporal data such as step length, step width, step time, foot rotation, cadence, and the duration of single support, swing, and double-stance phases of gait (refer to Table 1 for pairwise comparisons). Additionally, the analysis included the maximum vertical ground reaction forces (vGRFs) exerted on three geometrically divided plantar areas of the foot: the rear 30% (rearfoot), the middle 30% (midfoot), and the front 40% (forefoot). The maximum midfoot vGRF was the force exerted at the time point corresponding to the average of the time points at which maximum vGRFs were exerted on the rearfoot and forefoot. The anteroposterior and mediolateral position of the center of pressure (COP) was also determined based on the intersection point over time in the cyclogram [46]. This diagram is formed by connecting the trajectories of the COP from the forefoot of one side to the rearfoot of the contralateral side during the selected gait cycles [46]. Statistical analysis was conducted based on the average of the values obtained from all steps taken in each walking condition for each measured variable.

**Table 1.** Definition of the spatiotemporal parameters of gait recorded by the instrumented treadmill.

| Spatiotemporal Gait Parameters | Definition |
| --- | --- |
| Step length (cm) | Distance between the heel strike of one side of the body and the heel strike of the contralateral side |
| Step width (cm) | Distance between the right and left foot |
| Foot rotation (°) | Angle between the longitudinal axis of the foot and the direction of walking |
| Step time (s) | Time within a gait cycle between the heel strike of one side of the body and the heel strike of the contralateral side |
| Cadence (step/min) | Frequency of steps per unit time |
| Stance phase (s) | Time within a gait cycle when the foot is in contact with the ground |
| Swing phase (s) | Time within a gait cycle during which the foot is not in contact with the ground |
| Double-stance phase (s) | Time of the load response and the pre-swing phase collectively |

### 2.5. Statistical Analysis

The normality of the data distribution was assessed using the Shapiro–Wilk test and by visually inspecting Q-Q and box plot graphs. The paired *t*-test was employed to evaluate potential differences between the right and left sides concerning spatiotemporal and dynamic parameters measured in each leg. One-way repeated measures ANOVA was performed to detect possible differences between walking conditions for step length, width and time, cadence, and the duration of stance and swing phase. Differences between walking conditions (within-subjects factor) and foot areas (rearfoot, midfoot, and forefoot; between-subjects factor), as well as the walking condition-by-foot area interaction for vGRFs, were assessed using a mixed-design two-way ANOVA. The sphericity of the data was determined based on Mauchly's Test, where significant Greenhouse–Geisser correction was used. Pairwise comparisons were performed using the Bonferroni adjustment. The statistical analysis of the data was performed with SPSS 29.0 (IBM Corp, Armonk, NY, USA), while the significance level was set at the level of $p \leq 0.05$.

### 3. Results

As the differences between the right and left sides for spatiotemporal parameters and vGRFs measured bilaterally were not significant, the average of the values recorded on both sides was used in the statistical analysis performed in the present study.

### 3.1. Spatiotemporal Gait Parameters

Statistical analysis revealed significant differences between walking conditions for step length (F = 752.933, $p \leq 0.001$, partial $\eta^2$ = 0.960). Post-hoc comparisons indicated that,

compared to level walking, step length decreased as uphill and downhill walking became more demanding. Significant differences between walking conditions were found for step width (F = 57.802, $p \leq 0.001$, partial $\eta^2$ = 0.651). Compared to level walking, step width increased progressively during downhill walking, while only minor and non-significant changes were observed during uphill walking. There were significant differences in foot rotation across the different walking conditions (F = 33.199, $p \leq 0.001$, partial $\eta^2$ = 0.517). Post-hoc analysis revealed a significant increase in foot rotation during uphill walking compared to level walking. Additionally, foot rotation was lower in downhill walking, with significant differences observed only when comparing level walking to walking at a −10% slope and a speed of 5.0 km h$^{-1}$ (refer to Table 2 for pairwise comparisons).

Statistical analysis showed significant differences between walking conditions for step time (F = 362.858, $p \leq 0.001$, partial $\eta^2$ = 0.921) and cadence (F = 414.221, $p \leq 0.001$, partial $\eta^2$ = 0.930). Pairwise comparisons revealed that, in uphill walking conditions, step time significantly increased, and cadence decreased compared to level walking. Conversely, during downhill walking, step time significantly decreased, and cadence increased (refer to Table 2 for pairwise comparisons).

**Table 2.** Means ± standard deviations of spatiotemporal gait parameters.

| Spatiotemporal Gait Parameters | Walking Conditions (Slope/Speed) | | | | |
|---|---|---|---|---|---|
| | **A** | **B** | **C** | **D** | **E** |
| | −20% 3.5 km h$^{-1}$ | −10% 5.0 km h$^{-1}$ | 0% 5.0 km h$^{-1}$ | +10% 3.5 km h$^{-1}$ | +20% 2.5 km h$^{-1}$ |
| Step length (cm) | 46.9 ± 3.2 | 65.3 ± 3.6 [a] | 68.7 ± 3.4 [b] | 57.8 ± 3.9 [c] | 46.0 ± 4.1 [d] |
| Step width (cm) | 12.4 ± 2.3 | 11.7 ± 2.0 [e] | 9.3 ± 1.8 [b] | 9.3 ± 2.4 [b] | 10.0 ± 2.4 [f] |
| Foot rotation (°) | 5.5 ± 3.5 | 5.0 ± 3.3 | 6.0 ± 4.0 [g] | 7.1 ± 4.6 [c] | 8.7 ± 5.0 [h] |
| Step time (s) | 0.48 ± 0.03 | 0.47 ± 0.03 [i] | 0.49 ± 0.02 [j] | 0.59 ± 0.04 [c] | 0.67 ± 0.06 [h] |
| Cadence (step/min) | 125.2 ± 9.2 | 128.4 ± 7.3 [k] | 121.9 ± 6.0 [j] | 101.7 ± 7.1 [c] | 91.1 ± 8.3 [d] |

[a] Significant difference (SD) compared to A walking condition ($p \leq 0.001$); [b] SD compared to A and B walking condition ($p \leq 0.001$); [c] SD compared to A, B, and C walking condition ($p \leq 0.001$); [d] SD compared to B, C, and D walking condition ($p \leq 0.001$); [e] SD compared to A walking condition ($p \leq 0.01$); [f] SD compared to A, B ($p \leq 0.001$) and D ($p \leq 0.01$) walking condition; [g] SD compared to B walking condition ($p \leq 0.05$), and −10% slope ($p \leq 0.05$); [h] SD compared to A, B, C, and D walking condition ($p \leq 0.001$); [i] SD compared to A waling condition ($p \leq 0.01$); [j] SD compared to B waling condition ($p \leq 0.001$); [k] SD compared to A waling condition ($p \leq 0.05$).

Significant differences were observed for the stance phase (F = 505.472, $p \leq 0.001$, partial $\eta^2$ = 0.942), swing phase (F = 123.351, $p \leq 0.001$, partial $\eta^2$ = 0.799), and double-stance phase of gait (F = 750.943, $p \leq 0.001$, partial $\eta^2$ = 0.960) between the various walking conditions. Stance, swing, and double-stance durations increased in the uphill walking conditions and slightly decreased in downhill walking conditions compared to level walking (refer to Table 3 for pairwise comparisons).

**Table 3.** Means ± standard deviations of the duration of gait phases.

| Gait Phases | Walking Conditions (Slope/Speed) | | | | |
|---|---|---|---|---|---|
| | **A** | **B** | **C** | **D** | **E** |
| | −20% 3.5 km h$^{-1}$ | −10% 5.0 km h$^{-1}$ | 0% 5.0 km h$^{-1}$ | +10% 3.5 km h$^{-1}$ | +20% 2.5 km h$^{-1}$ |
| Stance phase (s) | 0.60 ± 0.05 | 0.57 ± 0.04 [a] | 0.61 ± 0.04 [b] | 0.76 ± 0.06 [c] | 0.89 ± 0.08 [d] |
| Swing phase (s) | 0.36 ± 0.02 | 0.37 ± 0.02 | 0.38 ± 0.02 [e] | 0.43 ± 0.03 [c] | 0.44 ± 0.04 [f] |
| Double-stance phase (s) | 0.24 ± 0.03 | 0.20 ± 0.03 [a] | 0.23 ± 0.03 [g] | 0.34 ± 0.04 [c] | 0.45 ± 0.05 [d] |

[a] Significant difference (SD) compared to A walking condition ($p \leq 0.001$); [b] SD compared to B walking condition ($p \leq 0.001$); [c] SD compared to A, B and C walking condition ($p \leq 0.001$); [d] SD compared to A, B, C and D walking condition ($p \leq 0.001$); [e] SD compared to A and B walking condition ($p \leq 0.001$); [f] SD compared to A, B, C ($p \leq 0.001$) and D walking condition ($p \leq 0.01$); [g] SD compared to A ($p \leq 0.05$) and B walking condition ($p \leq 0.001$).

### 3.2. Vertical Ground Reaction Forces

The results of the present study revealed significant main effects of walking conditions (F = 109.550, $p \leq 0.001$, partial $\eta^2$ = 0.541) and plantar foot areas (F = 300.107, $p \leq 0.001$, partial $\eta^2$ = 0.866) for the vGRFs. Additionally, a significant interaction was observed between walking conditions and plantar foot areas (F = 93.063, $p \leq 0.001$, partial $\eta^2$ = 0.667). Vertical ground reaction forces were significantly decreased on the rearfoot and forefoot plantar areas while remaining unchanged on the midfoot plantar area during both uphill and downhill walking conditions compared to level walking (refer to Figure 2 for pairwise comparisons).

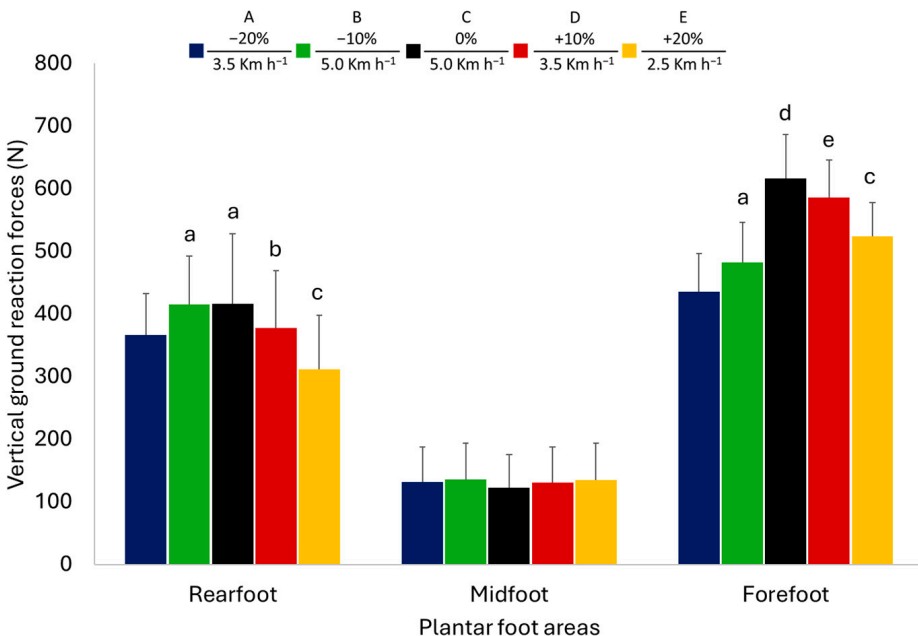

**Figure 2.** Maximum vertical ground reaction forces exerted on rearfoot, midfoot and forefoot during uphill (+) and downhill (−) walking at speeds determined by Tobler's hiking function. [a] Significant difference (SD) compared to A walking condition ($p \leq 0.001$); [b] SD compared to B and C walking condition ($p \leq 0.001$); [c] SD compared to A, B, C and D walking condition ($p \leq 0.001$); [d] SD compared to A and B walking condition ($p \leq 0.001$); [e] SD compared to A, B and C walking condition ($p \leq 0.001$).

### 3.3. COP Position in the Anteroposterior and Mediolateral Direction

Significant were the differences between walking conditions regarding the position of COP in the anteroposterior (F = 218.384, $p \leq 0.001$, partial $\eta^2$ = 0.876) and mediolateral direction (F = 5.075, $p \leq 0.01$, partial $\eta^2$ = 0.141). COP shifted more anteriorly during uphill walking and more posteriorly during downhill walking compared to level walking. COP shifted more mediolaterally during uphill and downhill walking compared to level walking (refer to Figure 3 for pairwise comparisons).

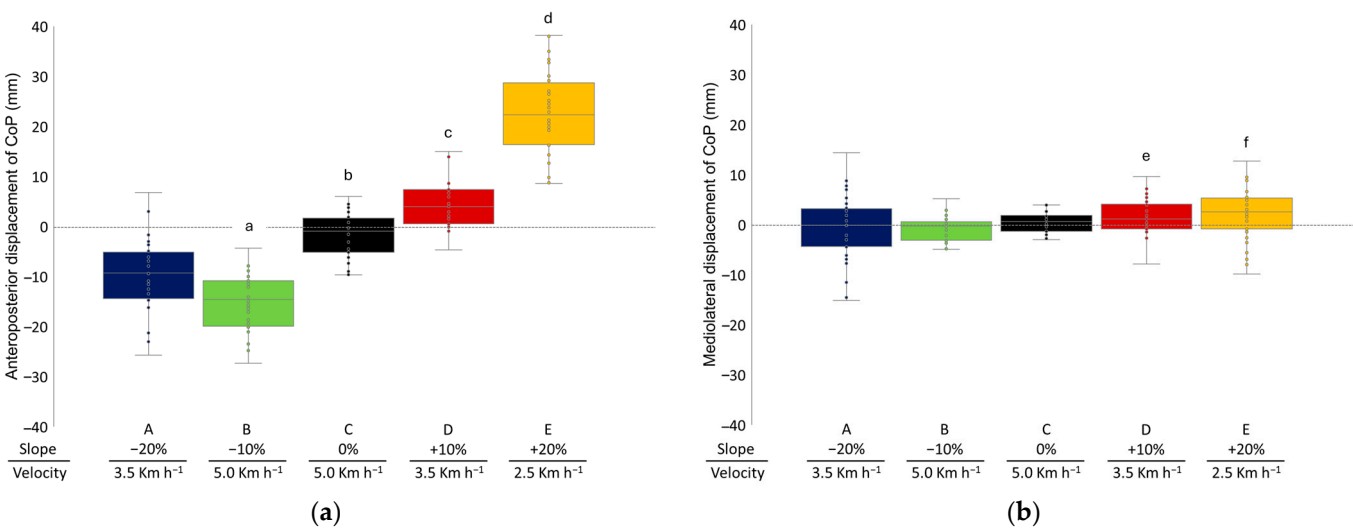

(a)                                                                                    (b)

**Figure 3.** Box plots for center of pressure (COP) positions in (**a**) anteroposterior and (**b**) mediolateral directions during uphill (+) and downhill (−) walking at speeds determined by Tobler's hiking function. [a] Significant difference (SD) compared to A walking condition ($p < 0.001$); [b] SD compared to A and B walking condition ($p < 0.001$); [c] SD compared to A, B and C walking condition ($p < 0.001$); [d] SD compared to A, B, C and D walking condition ($p < 0.001$); [e] SD compared to B walking condition ($p < 0.05$); [f] SD compared to B ($p < 0.01$) and C ($p < 0.05$) walking condition.

## 4. Discussion

### 4.1. Spatiotemporal Parameters

Our findings revealed that walking uphill on progressively steeper slopes and at lower speeds was associated with significantly shorter steps and more outward foot rotation compared to level walking. Downhill walking, on the other hand, resulted in shorter and markedly wider steps compared to level walking (Figure 4).

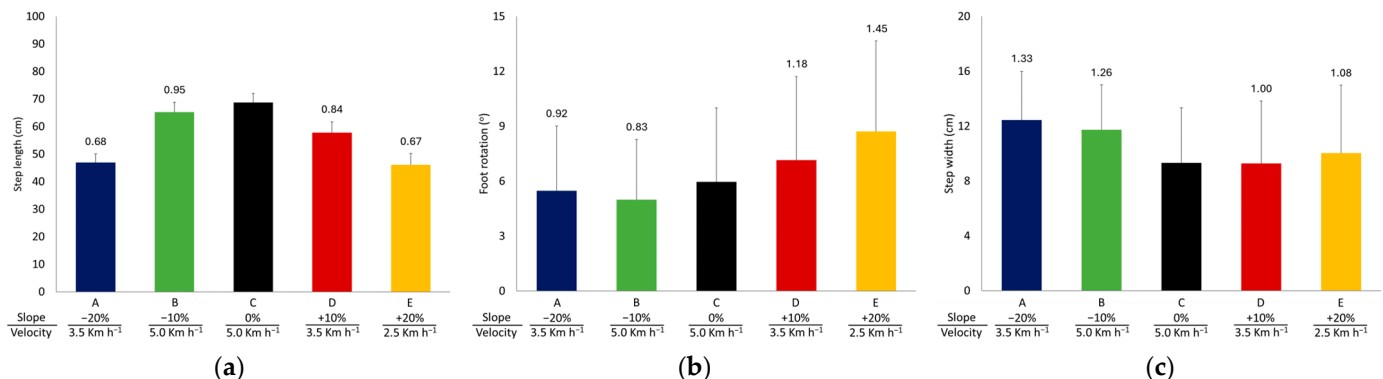

(a)                                                (b)                                                (c)

**Figure 4.** Mean and standard deviation (error bars) of (**a**) step length, (**b**) foot rotation, and (**c**) step width under different walking conditions. The change in each spatial parameter for the uphill and downhill walking conditions is presented as a multiple of the corresponding mean spatial parameter in the C walking condition.

In terms of temporal parameters, step time increased, and cadence decreased during uphill walking conditions, whereas the opposite occurred during downhill walking conditions compared to level walking (Figure 5).

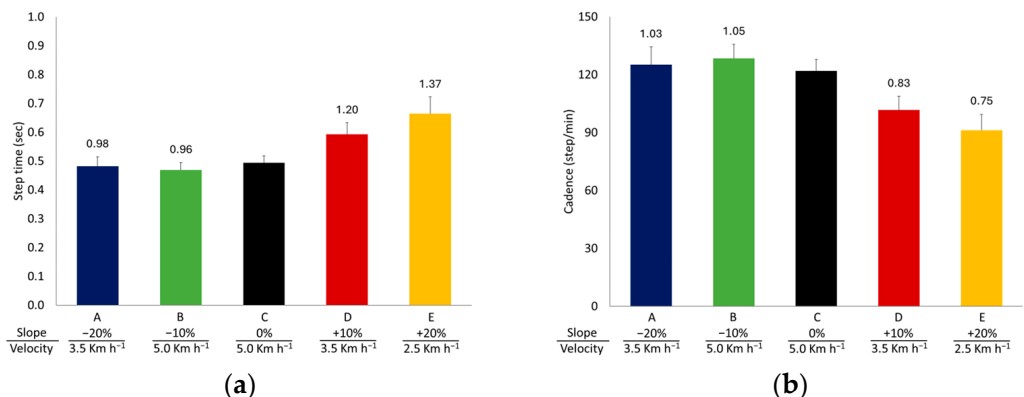

(**a**)          (**b**)

**Figure 5.** Mean and standard deviation (error bars) of (**a**) step time and (**b**) cadence under different walking conditions. The change in each temporal parameter for the uphill and downhill walking conditions is presented as a multiple of the corresponding mean temporal parameter in the C walking condition.

The duration of the stance and swing phase, as well as the duration of the double-stance phase, increased during uphill walking but slightly decreased during downhill walking compared to level walking conditions (Figure 6).

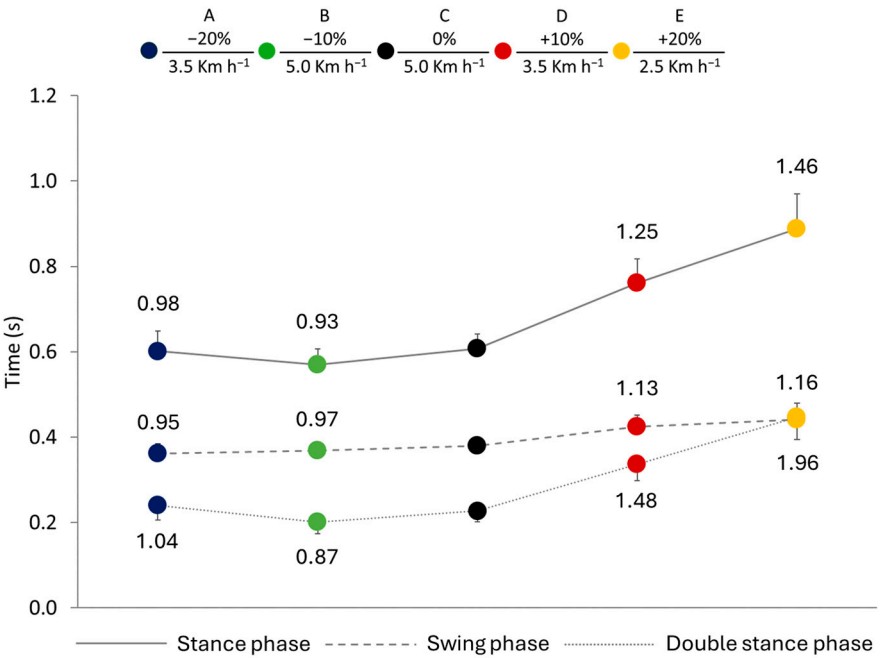

**Figure 6.** Mean and standard deviation (error bars) of stance, swing, and double-stance phase duration under different walking conditions. The change in the duration of each gait phase for the uphill and downhill walking conditions is presented as a multiple of the corresponding mean gait phase duration in the C walking condition.

The differences in spatiotemporal parameters of gait observed during uphill and downhill walking could be attributed to mechanical/kinematic responses, physiologic adjustments, and sensory modifications that enable participants to cope with gravitational resistance during uphill walking and to manage excessive impact forces during downhill walking. Previous studies have shown that during uphill walking from 0 to 10%, there is a progressive forward tilt of the trunk and pelvis, accompanied by an increasingly flexed posture of the hip, knee, and ankle at initial foot contact [15,47]. Conversely, downhill walking from 0 to −10% induces a progressive backward tilt of the trunk and pelvis, along with a decreasingly flexed posture of the hip at initial foot contact, as well as an increase in

knee flexion during weight acceptance and late stance [15,47]. These kinematic responses may be partly responsible for the changes in step length, which, according to some researchers, explain its increase during uphill walking and its decrease during downhill walking [15,47]. However, these findings contradict the results of the present study, as well as other research, which demonstrated a reduction in step length during both uphill and downhill walking [16,17]. Differences between studies regarding step length may result from the speed implemented for walking on various slopes. The studies demonstrating increased step/stride length during uphill walking maintained a preferred or fixed speed that remained constant across all slopes studied, therefore forcing participants to face more challenging conditions compared to level walking [15,47]. In contrast, the speed chosen for uphill walking in our study progressively decreased as the gradients became steeper, allowing participants to cope with gravitational resistance by propelling their bodies forward with smaller steps.

Another factor that could have affected step length during uphill and downhill walking is the concurrent generation of frictional force. Research has shown that the friction requirements at heel strike are decreased, while those at toe-off are increased during uphill walking compared to level walking [48]. When walking uphill, the resistance imposed by the inclined surface amplifies frictional resistance, demanding greater effort to push off the ground during toe-off [7]. This heightened resistance makes it more challenging to generate forward propulsion. Consequently, individuals may find it difficult to push off with the same force as they would on a flat surface, prompting them to shorten their stride length. Furthermore, the gravitational force that accelerates the body during downhill walking amplifies the frictional demand, especially at the heel strike, where the foot contacts the ground [48]. A decrease in step length might mitigate the frictional demand since shorter steps entail less force exerted upon heel strike [48].

Besides changes in step length, we observed an increase in outward rotation of the foot, especially pronounced on steeper uphill slopes, and a widening of step width, particularly noticeable on increasingly steep downhill slopes. Participants in our study may have rotated their feet outward to prevent tripping while improving foot-ground contact to generate a strong push against gravitational forces. This response likely stemmed from limitations of adequate hip and knee flexion, which is expected due to both the anterior tilt of the trunk and pelvis, which is expected to occur during uphill walking [15,47]. Moreover, it may result from insufficient dorsiflexion of the ankle due to the limitations it presents, especially in athletic individuals like the participants in the present study [49]. The outward rotation of the foot also may be achieved by peroneus longus, whose activation tends to increase during uphill walking, by functioning as an evertor and abductor of the foot, thus potentially overpowering the counter activation of the tibialis anterior, which also increases under similar walking conditions [50]. The outward rotation of the foot might have been further facilitated by heightened activation of the biceps femoris, a biarticular muscle with an insertion on the head of the fibula, allowing it to serve as an external rotator of the tibia and leg. This muscle's increased activation is attributable to its role as a hip and knee extensor, aiding in elevating the body's center of mass during uphill walking [51].

The increased step width during downhill walking likely stemmed from increased lateral instability, as demonstrated by the heightened shift of COP in the mediolateral direction, representing a strategic biomechanical adaptation aimed at widening the base of support, crucial for navigating steep descents where balance becomes paramount. Under these walking conditions, body stability could have been compromised by the inherent instability of the ankle joint due to its plantar flexion position when the foot contacts the declined ground. In this position, the ankle joint is in an open-packed position characterized by decreased bony stability and ligamentous tension [52]. Another factor that may have contributed to the increase in step width during downhill walking could be the disturbed proprioception and associated joint stability induced by the eccentric contraction of lower limb muscles governing such activities [53]. Research has shown that eccentric contractions disturb both the muscle spindle and Golgi tendon organs' proprioception [54,55], poten-

tially disrupting body balance, as both sensory inputs are believed to contribute to joint stability by modulating the stiffness of muscles controlling the joint [56]. Other studies have also demonstrated that a 30-min downhill walking [11] and as low as 20 reciprocal isokinetic maximum concentric and eccentric contractions at an angular velocity of 180 degrees per second [57] can deteriorate knee joint-position sense, indicating that both prolonged eccentric low load activity and local loading might compromise joint proprioception, thus ultimately affecting joint stability. Moreover, widening the step during downward walking may have served to prevent overstriding, enabling the impact of each step to be more uniformly distributed over a larger surface area. This could potentially relieve stress on joints and muscles, particularly in the lower extremities, thus reducing the risk of injury (e.g., falls) [58,59].

There is a consensus among the present and other published studies regarding temporal characteristics such as step, stride, stance, and swing time, with all of them showing an increase during uphill and a decrease in downhill walking [15,17,47]. Consistent were also our findings with findings reported elsewhere regarding the decreased cadence during uphill and increased cadence during downhill walking, albeit not always significant compared to level walking [16,47]. Given that all participants in the present study were students of similar age, musculoskeletal health, and fitness level, and considering that they walked without shoes and had the same period of adaptation to the treadmill, any changes in temporal parameters are expected to have resulted from the variation in speed selection for walking on the different slopes. The progressively lower speeds determined for walking at progressively increasing slopes resulted in gradually increasing step time and gait cycle duration, ultimately decreasing cadence. Fukuchi et al. [60], after conducting an extensive and systematic review of the existing literature, reported that lower cadence and greater stance duration should be expected at slower speeds during level walking. Muscle activation that increases during uphill walking [7,18] is likely to have further contributed to the observed changes in the temporal parameters studied, as the reduced walking speed may enable the production of the force necessary to propel the body. During downhill walking, despite the speed being the same or slower than the level walking speed, participants exhibited shorter step times and gait cycle durations, leading to an increase in cadence. This adjustment may have been necessary for participants to synchronize their pace with the moving treadmill belt as it passed beneath their feet. These responses are likely attributed to gravity-assisted propulsion, combined with a reduced necessity for muscle activation [7,18].

### 4.2. Vertical Ground Reaction Forces

The study's findings revealed lower vGRFs on both the rearfoot and forefoot at all slopes except for the −10% slope when compared to walking on a 0% slope. There were no significant differences between vGRFs generated on the midfoot during sloping gait compared to level gait. A progressive decrease with increasing gradients was also observed in vGRFs during both uphill and downhill walking, affecting both the rearfoot and forefoot. However, forefoot forces during uphill walking were greater than those during downhill walking, while the opposite trend was observed in the rearfoot (Figure 7).

The vGRFs during uphill and downhill walking were generally lower compared to level walking, possibly due to biomechanical adaptations made by the body to overcome or counteract the applied gravitational resistance, respectively. Overcoming gravity by adopting a forward-tilted trunk and pelvis posture [15,47], as indicated in our study by shifting the center of mass slightly forward of the center of the foot, combined with the shorter and slower steps taken during uphill walking, resulted in changing the orientation of the ground reaction forces. In this case, the vertical component of the ground reaction force, which is perpendicular to the treadmill's surface, decreases as the uphill slope increases compared to level walking because the gravitational force is partially offset by the slope. Conversely, the net effect of the gravitational force assisting the body's descent during downhill walking and the treadmill's slope seemed to have little influence on the vertical component of the ground reaction force, except when encountering a 20% slope.

Moreover, at this slope, downhill walking may promote forward movement, therefore diminishing the necessity for active propulsion and subsequently lowering vGRFs in comparison to level walking.

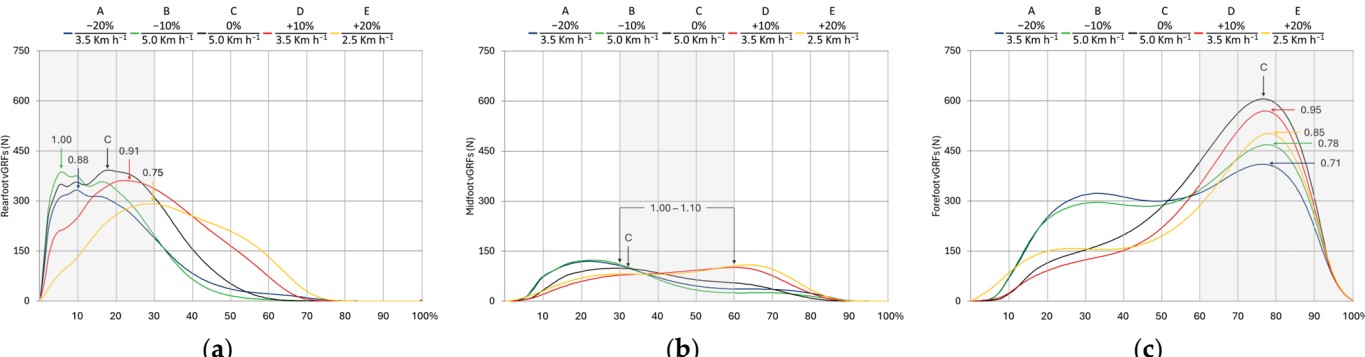

**Figure 7.** Force curves showing the mean vGRFs progression in (**a**) rearfoot, (**b**) midfoot, and (**c**) forefoot zones (i.e., as long as load occurs in the corresponding zone) over the gait cycle under various walking conditions. The changes in mean vGRFs for uphill and downhill walking conditions are presented as multiples of the maximum vGRF elicited in the corresponding zone (shaded areas) in the C walking condition.

Another factor that may have influenced vGRFs was the speeds selected for uphill and downhill walking, which, according to Tobler's hiking function and empirical data [23,61], decrease as the gradient of the treadmill's surface increases, except for the 10% downhill walking, where speed remains the same as that of level walking. Early studies have shown that vGRFs increased as speed increased during level walking [62,63]. Koo et al. [22], in a recent study, found that peak pressures in both the forefoot and rearfoot decrease when walking on uphill and downhill slopes of 4 and 8 degrees at lower speeds compared to higher speeds. Other researchers [64], who examined ground reaction force profiles during inclined running at iso-efficiency speeds—speeds maintaining the same metabolic intensity as level running—found that running at 4% and 8% inclinations, with lower speeds compared to level running, resulted in significantly reduced peak vertical ground reaction forces as the treadmill incline increased.

Alterations in muscle activation, which occur as the foot strike angle changes with variations in ground slope, may contribute to the differences obtained in vGRFs compared to level walking, as well as the redistribution of vGRFs between the plantar areas of the foot in various walking conditions. The heightened need for the body to generate forward propulsion to overcome gravity and friction during uphill walking leads to increased activity of the hip and knee extensors as well as the plantar flexors as the gradient of the surface increases [7,18]. This increased activity, in turn, may induce significantly greater vGRF on the forefoot compared to the rearfoot, as revealed in the present and previous studies [21]. However, these forces may have decreased as the slope increased, as uphill walking was selected to be performed at slower speeds, requiring less muscle activation [7]. In contrast, the biomechanical adjustments made by the body during downhill walking did not necessitate increased activation by the leg muscles [7,18], as the propelling forward of the body is assisted by gravity. The activation of the knee extensors, however, which is likely increased to control the descent and absorb the gravitational shock generated during heel strike [7,18], may eventually increase vGRFs on the rearfoot compared to the forefoot, a finding supported by other researchers [20]. Moreover, the progressive decrease in vGRFs observed in both the rearfoot and the forefoot was likely due to reduced muscle activation that typically occurs when downhill walking is performed at lower speeds compared to faster speeds [7,18], such as those selected in our study for walking on −20% (3.5 km h$^{-1}$) and −10% and 0% slopes (5.0 km h$^{-1}$).

### 4.3. COP Position in Anteroposterior and Mediolateral Direction

Our findings revealed an anterior shift of the COP during uphill walking compared to level walking, with this shift becoming more pronounced as the slope increased. Conversely, during downhill walking, the COP shifted backward compared to level walking, with this backward shift being more prominent at a −10% slope. Furthermore, the mediolateral shift of COP increased during uphill and downhill walking compared to level walking, and this shift tended to increase, although not always significantly, as the slope became steeper (Figure 8). These findings are consistent with previously reported data that have shown a greater shift in the anteroposterior direction and mediolateral excursions of the COP at uphill inclinations of 15 and 25 degrees [65].

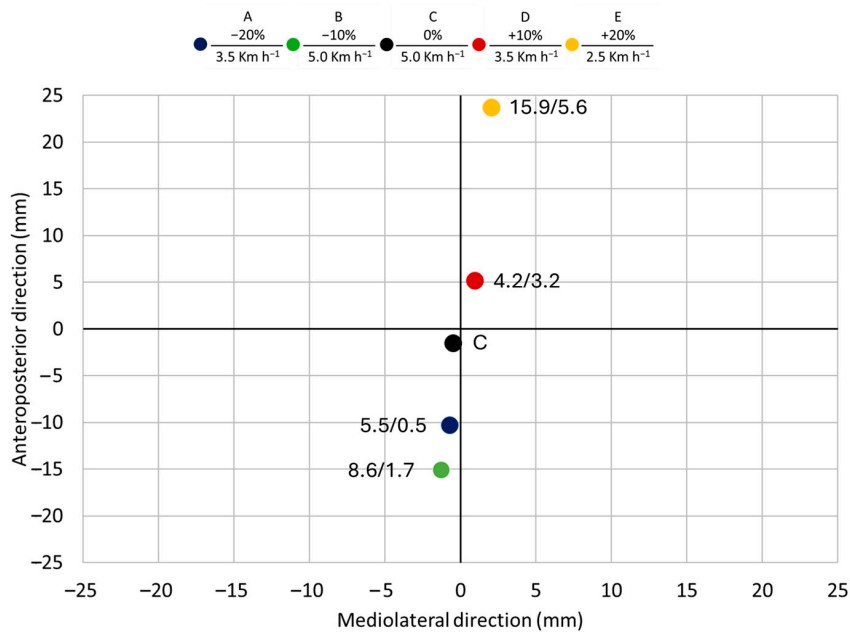

**Figure 8.** Mean COP position in the anteroposterior and mediolateral directions under various walking conditions. The mean displacement of COP in uphill and downhill walking conditions is presented as multiples of the mean COP position in the anteroposterior and mediolateral directions determined in the C walking condition.

The increased anteroposterior displacement of COP during uphill walking can be attributed to participants' postural adjustments, necessitating them to lean forward to counteract gravitational forces and sustain balance [15]. Leroux et al. [15] hypothesized that by inclining the trunk forward, the center of mass would shift slightly ahead of the foot's center, causing gravity to propel the body forward relative to the center of support. By shifting the center of mass forward, individuals can effectively distribute their body weight over their feet and propel themselves upward against the force of gravity. Given the method of determining the COP in this study (by connecting the trajectories of the COP from the forefoot of one side to the rearfoot of the contralateral side during selected gait cycles), our data support this hypothesis, indicating that vGRFs are exerted more anteriorly on the plantar foot areas. In contrast, when descending, individuals must lean backward to control their speed and prevent falling [15], leading to a more backward application of vGRFs and, ultimately, a posterior shift of the COP. Moreover, the mediolateral displacement of the COP tended to increase, although not always significantly, with increasing slope. This indicates that individuals may need to make greater lateral adjustments to maintain balance and stability on steeper slopes while navigating uneven terrain and adapting to changes in slope. In our study, these adjustments were manifested with increased foot rotation during uphill walking and increased step width during downhill walking, both indicating an expansion of the base of support.

### 4.4. Clinical Implications

The findings of our research aim to provide information to people from various domains, such as long-distance walkers, by offering insights into the decision-making process in route selection and strategies that consider individual characteristics while accommodating special needs and travel preferences [1]. Combining individuals' specific needs with the changing walking conditions along a route may enable walkers to anticipate the challenges posed by uphill and downhill sections, allowing them to adjust their pace and conserve energy accordingly. Public health providers can also benefit from understanding the potential effects of walking and advocating for it as a therapeutic intervention to enhance overall physical health. Indeed, studies have demonstrated that the efforts required to overcome the challenges faced during uphill and downhill walking are correlated with enhanced cardiovascular health [3,4], increased muscle activation [7,8], and the management of chronic conditions like diabetes [6], obesity [5], COPD [13] and osteoarthritis [66]. In addition, healthcare professionals, including physical therapists and rehabilitation specialists, can utilize spatiotemporal parameters to adjust and optimize rehabilitation programs [28,29] by improving certain functional abilities (e.g., muscle strength) or identifying specific gait abnormalities and tracking changes over time. Understanding ground reaction force and GRF patterns may also facilitate the implementation of intervention programs to improve weight distribution and reduce joint stress, thus alleviating asymmetries or excessive joint loading. This would be particularly important in the prevention of musculoskeletal injuries resulting from frequent and prolonged walking on various surfaces, such as blisters [67] and ankle sprains [68]. Likewise, COP data may allow for the design of exercises that specifically target balance deficits and enhance overall stability. Understanding the impact of gait mechanics and ground reaction forces can assist footwear designers in developing shoes with suitable cushioning and support to minimize impact forces and reduce the risk of injury during activities like hiking [27]. Finally, such findings may assist government agencies and organizations responsible for the management of natural parks and open spaces in improving and maintaining existing footpaths [69] and designing new ones to ensure public safety and enjoyment [26]. Knowing how people move on different terrains and slopes can help (i) identify areas prone to accidents or hazardous conditions, (ii) improve accessibility and comfort for all visitors, (iii) identify high-traffic areas and implement measures to minimize erosion, protect sensitive ecosystems, conserve biodiversity and manage resources such as maintenance, signage and infrastructure upgrades, and (iv) create pleasant outdoor spaces experiences for visitors while reducing congestion and conflicts.

### 4.5. Study Limitations

A limitation of our study was that walking on various incline conditions was conducted without participants wearing shoes. It has been reported that barefoot walking affects spatiotemporal and kinetic parameters of gait by reducing step/stride length, increasing cadence, and reducing peak vertical ground reaction force at initial contact compared to wearing common footwear [70]. Participants in our study decided to walk barefoot based on several factors. First, the belief that force transmission would be more accurate, given that the foot could move freely without interference from the force absorption or structure of the shoe sole, thus reducing artifacts or measurement inaccuracies. Second, there is an expectation of more direct sensory feedback to the central nervous system regarding foot position and sensation facilitated by proprioceptors located on its plantar surface. Thirdly, the unavailability of a specific shoe type could mitigate potential distortion and variability arising from different shoe sole types in spatiotemporal gait parameters and/or ground reaction force magnitudes. Eventually, this more natural foot functioning could yield valuable insights for various professionals, including shoe designers, as their designs necessitate a comprehensive understanding of plantar force variations under the studied conditions. Given that footwear affects the kinematics and kinetics of gait both acutely and chronically [70], the results of this study should be interpreted with caution, as barefoot walking may have influenced the observed gait parameters in our study. Other limitations

related to sample characteristics and methodology may also limit the generalizability of our findings. The study focused on healthy young adults, potentially limiting the generalizability of the findings to other age groups or individuals with specific health conditions. Different age groups [71] and fitness levels might exhibit varied responses when walking uphill and downhill at the speeds examined. Furthermore, this study did not extensively explore individual differences that could influence gait responses. Factors such as individual biomechanics, adaptation to participants' treadmill gait, prior long-distance walking experience, psychological factors, or variations in participant motivation during the study could impact gait parameters and ground reaction forces differently compared to walking on natural terrain. The possibility that participants modified their behavior simply due to being observed (Hawthorne effect) may also have influenced their walking patterns [72]. This heightened awareness could result in altered behaviors that may not accurately reflect natural walking conditions. Finally, walking conditions implemented in this study might not fully capture the intricacies of hiking on varied outdoor landscapes, as the study is conducted on a treadmill, and the findings may not fully generalize to outdoor hiking conditions [33]. Treadmill walking lacks the variability and unpredictability associated with natural terrains, potentially limiting the ecological validity of the results.

## 5. Conclusions

Our findings revealed that speed-adjusted sloping gait results in spatiotemporal, vGRFs, and COP changes that are necessary to face the challenges faced during both uphill and downhill walking. During uphill walking, we observe a decrease in stride length and cadence alongside an increase in foot rotation, stride time, and durations of stance, swing, and double-stance phases. Concurrently, there is a progressive reduction in vGRFs and an anterior and lateral displacement of the COP as the surface slope increases. In contrast, downhill walking showed decreased step length, step time, and durations of stance, swing, and double-stance phases while exhibiting increased step width and cadence compared to level walking. Additionally, vGRFs decreased, and the COP shifted posteriorly and laterally as the slope descended. Understanding the impact of these factors will not only enhance our comprehension of the biomechanics of hiking but also inform strategies to optimize performance and mitigate potential musculoskeletal risks associated with hiking-related activities.

**Author Contributions:** Conceptualization, I.K. (Ioannis Kafetzakis) and D.M.; methodology, I.K. (Ioannis Kafetzakis) and D.M; validation, I.K. (Ioannis Kafetzakis) and I.K. (Ilias Konstantinou); formal analysis, I.K. (Ioannis Kafetzakis) and D.M.; investigation, I.K. (Ioannis Kafetzakis); resources, I.K. (Ioannis Kafetzakis) and I.K. (Ilias Konstantinou); data curation, I.K. (Ioannis Kafetzakis); writing—original draft preparation, I.K. (Ioannis Kafetzakis); writing—review and editing, D.M.; visualization, D.M.; supervision, I.K. (Ilias Konstantinou) and D.M.; project administration, I.K. (Ioannis Kafetzakis). All authors have read and agreed to the published version of the manuscript.

**Funding:** This research received no external funding.

**Institutional Review Board Statement:** The study was conducted according to the guidelines of the Declaration of Helsinki and approved by the Bioethics committee of the School of Physical Education and Sport Science of the National and Kapodistrian University of Athens (Reg. No 1338/15-12-2021).

**Informed Consent Statement:** Informed consent was obtained from all subjects involved in the study.

**Data Availability Statement:** The data presented in this study are available upon request from the corresponding author.

**Conflicts of Interest:** Author Ilias Konstantinou was employed by the company Serinth Ltd. The remaining authors declare that the research was conducted in the absence of any commercial or financial relationships that could be construed as a potential conflict of interest.

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
