# Peer review of "Effects of Hiking-Dependent Walking Speeds and Slopes on Spatiotemporal Gait Parameters and Ground Reaction Forces: A Treadmill-Based Analysis in Healthy Young Adults"

_applsci, doi:10.3390/app14114383_

Round 1

Reviewer 1 Report

Comments and Suggestions for Authors

Applied Sciences

Manuscript ID: applsci-2978041

Effects of Hiking-Depended Walking Speeds and Slopes on Spatiotemporal Gait Parameters and Ground Reaction Forces. A Treadmill Based Analysis in Healthy Young Adults

The study investigated the spatiotemporal parameters, ground reaction forces (vGRFs), and center of pressure (COP) displacements during speed-adjusted uphill and downhill walking. Findings showed that uphill walking led to shorter steps, increased foot rotation, longer step times, and greater stance/swing phase durations, with decreased vGRFs and anterior/lateral COP displacements. Downhill walking resulted in shorter steps, shorter step times, wider steps, and increased cadence, with decreased vGRFs and posterior/lateral COP displacements. Understanding these biomechanical changes can inform strategies to optimize hiking performance and reduce musculoskeletal risks associated with hiking-related activities. In general, the experiment is well discussed, and the analysis is clear. However, there are some points that the authors need to provide and/or explain and some issues that need to be addressed.

 ·       Abstract:

o   It could benefit from more specific quantitative results. For example, stating the percentage change in stride length or cadence would provide a clearer picture of the effects of hiking-dependent walking speeds and slopes.

o   The abstract briefly touches on the practical implications of the findings, but could expand on how these results contribute to the existing body of knowledge in biomechanics and hiking.

·       I believe that including a figure depicting the experimental setup, as well as a picture of the experiment itself, would greatly enhance the clarity and visual appeal of your study. These additions could provide readers with a clearer understanding of your experimental design and the conditions under which your data were collected.

·       Having a sketch could be beneficial to illustrate the spatiotemporal parameters, ground reaction forces, and center of pressure displacements during speed-adjusted sloping gait. Visual representation can help readers better understand the biomechanical responses discussed in the study. It could also enhance the clarity of the methodology and results, making the findings more accessible to a wider audience.

·       Table 1 is fine but having a figure showing the spatiotemporal parameters of gait recorded by the instrumented treadmill would be much more informative.

·       A curve would be more effective for showing the changes in stance, swing, and double stance durations across different walking conditions could be used to visually represent how these durations vary for each condition, making it easier for readers to compare the changes and understand the impact of the different slopes on gait parameters.

·       The authors did not consider individual differences or gender as important factors, which could impact gait responses. Could you consider incorporating an analysis of gender differences in gait patterns into your study? This addition could provide valuable insights into how gender impacts sloping gait biomechanics, enhancing the overall depth and relevance of your findings. If there is no notable difference, including this information would still be valuable.

·       Regarding Clinical implications:

o   The section could be strengthened by providing more specific details about how the findings can be applied in each domain. For example, how can the study help long-distance walkers make more informed route selections? How can it guide the design of rehabilitation programs for physical therapists?

o   While the section mentions various stakeholder groups, it does not delve into the specific needs or challenges faced by each group. Providing more tailored recommendations based on the findings of the study would enhance the practicality and relevance of the clinical implications.

Author Response

NA

Reviewer 2 Report

Comments and Suggestions for Authors

The text contains precise descriptions of changes in gait parameters and kinematic and kinetic responses during uphill and downhill walking. The study appears to have been conducted systematically, using appropriate measurement methods. The results may be useful in understanding how the human body reacts to different terrain conditions during hiking.

The authors appropriately described the limitations of the study, but did not precisely justify why they made certain decisions:

Why was a barefoot walking protocol adopted? I understand that different types of shoes significantly affect ground reaction forces and gait parameters. However, in natural hiking conditions, everyone wears shoes. It would have been better to use one type of footwear in the study, such as athletic shoes or hiking boots. With significant impact forces, such as when descending quickly, the lack of shoe sole cushioning at the heel will force a different footwork and disrupt gait parameters that could be maintained while walking in shoes. Please clarify this decision in the study method description.

How and when was maximum heart rate determined during the study, and how is heart rate measured during the test (with what device)? This information is missing from the research protocol.

Arbitrarily assuming an average speed of 5 km/h for all participants is not a good idea. Speed depends on the individual characteristics of the participant, such as height, weight, and lower limb proportions. It would have been better to use as a reference point the natural, unforced walking speed measured in the field on a flat track, rather than on a mechanical treadmill. Please address this suggestion in the methods section.

To specify the influence of incline and speed on gait parameters, it would be necessary to examine walking at the same speeds on different inclines.

In biomechanical studies of gait, speed is typically reported in m/s.

Author Response

NA
